

# Metabolic marker gene mining provides insight in global *mcrA* diversity and, coupled with targeted genome reconstruction, sheds further light on metabolic potential of the *Methanomassiliicoccales*

Daan R. Speth and Victoria J. Orphan

Division of Geological and Planetary Sciences, California Institute of Technology, Pasadena, CA, United States of America

Corresponding authors
Daan R. Speth, dspeth@caltech.edu
Victoria J. Orphan,
vorphan@gps.caltech.edu

## ABSTRACT

Over the past years, metagenomics has revolutionized our view of microbial diversity. Moreover, extracting near-complete genomes from metagenomes has led to the discovery of known metabolic traits in unsuspected lineages. Genome-resolved metagenomics relies on assembly of the sequencing reads and subsequent binning of assembled contigs, which might be hampered by strain heterogeneity or low abundance of a target organism. Here we present a complementary approach, metagenome marker gene mining, and use it to assess the global diversity of archaeal methane metabolism through the *mcrA* gene. To this end, we have screened 18,465 metagenomes for the presence of reads matching a database representative of all known mcrA proteins and reconstructed gene sequences from the matching reads. We use our mcrA dataset to assess the environmental distribution of the *Methanomassiliicoccales* and reconstruct and analyze a draft genome belonging to the 'Lake Pavin cluster', an uncultivated environmental clade of the *Methanomassiliicoccales*. Analysis of the 'Lake Pavin cluster' draft genome suggests that this organism has a more restricted capacity for hydrogenotrophic methylotrophic methanogenesis than previously studied *Methanomassiliicoccales*, with only genes for growth on methanol present. However, the presence of the soluble subunits of methyltetrahydromethanopterin:CoM methyltransferase (*mtrAH*) provide hypothetical pathways for methanol fermentation, and aceticlastic methanogenesis that await experimental verification. Thus, we show that marker gene mining can enhance the discovery power of metagenomics, by identifying novel lineages and aiding selection of targets for in-depth analyses. Marker gene mining is less sensitive to strain heterogeneity and has a lower abundance threshold than genome-resolved metagenomics, as it only requires short contigs and there is no binning step. Additionally, it is computationally cheaper than genome resolved metagenomics, since only a small subset of reads needs to be assembled. It is therefore a suitable approach to extract knowledge from the many publicly available sequencing projects.

# INTRODUCTION

Genome resolved metagenomics is allowing unprecedented, primer independent, insight in the diversity of the microbial world (*Tyson et al., 2004*; *Hug et al., 2016*). In addition to the window into microbial diversity that metagenomics sequencing offers, it also provides clues for the metabolism of the organisms observed (*Tyson et al., 2004*). More precisely, based on the presence (or absence) of homologs of previously studied genes, an educated guess of the metabolism of an organism can be made. This has amongst others, led to the recent discovery of complete ammonium oxidation (comammox) in a single organism (*Van Kessel et al., 2015*; *Daims et al., 2015*) and provided evidence for archaeal methane metabolism outside of the *Euryarchaeota* (*Evans et al., 2015*; *Vanwonterghem et al., 2016*).

Other major advances in our understanding of the diversity of archaeal methane metabolism have come from cultivation studies, including the culturing and enrichment of members of the 7th *Euryarchaeal* order of methanogens, the *Methanomassiliicoccales* (*Dridi et al., 2012*; *Borrel et al., 2012*; *Iino et al., 2013*; *Borrel et al., 2013a*), and the recent culturing of halophilic methanogens from Siberian soda lakes (*Sorokin et al., 2017*). The latter group seems to be restricted to highly saline environments, whereas environmental sequencing indicates that the *Methanomassiliicoccales* are widely distributed, occurring in habitats ranging from animal guts to wetlands and wastewater treatment (*Großkopf, Stubner & Liesack, 1998*; *Tajima et al., 2001*; *Wright et al., 2004*; *Iino et al., 2013*; *Söllinger et al., 2016*). Indeed, a recent large-scale effort to bin genomes from environmental metagenome data recovered 66 *Methanomassiliicoccales* genomes (*Parks et al., 2017*), further supporting the environmental relevance of this order. It has recently been proposed that the *Methanomassiliicoccales* should be divided into an environmental clade and a gastrointestinal tract (GIT) clade (*Söllinger et al., 2016*). In addition, a 'Lake Pavin' clade, named after the site where it was first detected, was previously proposed based on analysis of environmental 16S ribosomal RNA gene sequencing (*Borrel et al., 2013b*). Another recent study suggested the existence of *Methanomassiliicoccales* in marine sediments, based on the presence of butanetriol dibiphytanyl glycerol tetraether (BDGT) lipids in *Methanomassiliicoccus luminyensis*, and the detection of these lipids in marine sediments (*Becker et al., 2016*), but the specificity of this biomarker is unclear.

The recent advances in our understanding of the diversity of archaeal methane metabolism raise the question whether additional novel diversity exists within previously sequenced metagenomic datasets, and whether the environmental importance and diversity of understudied clades can be further illuminated. In the examples of metagenomics enabled discovery discussed above, homologs of marker genes known to be diagnostic for methane metabolism were discovered in metagenome assembled genomes (MAGs). This requires the assembly of the raw sequencing reads, and subsequent binning of the assembled contigs into draft genomes (*Dick et al., 2009*; *Thomas, Gilbert & Meyer, 2012*). Assembly is computationally expensive on large datasets, and strain diversity within a sample can result in highly fragmented assemblies (*Thomas, Gilbert & Meyer, 2012*). Automated binning has improved dramatically in recent years, but this process often still requires substantial time-consuming manual curation (*Albertsen et al., 2013*; *Delmont et al., 2017*).

Alternatively, the diversity of organisms capable of a metabolic process can be assessed using PCR-based screening of environmental samples. However, PCR-based analyses are sensitive to primer bias, and therefore unlikely to yield highly divergent gene sequences. In addition, the amplification of a single metabolic gene makes elucidation of the taxonomic affiliation of the organism containing the gene difficult, although for some well-studied marker genes the coupling between gene and organism phylogeny has been documented.

Directly mining metagenomic reads for marker genes, and subsequently reconstructing the full-length gene sequence, combines some of the advantages of both of these strategies, while minimizing the disadvantages. Using a curated database, reads can be confidently assigned to a gene of interest, with false positive removal using a BLAST Score Ratio (BSR) (*Rasko, Myers & Ravel, 2005*). If a divergent variant of a gene of interest is retrieved using this approach, the genome containing this gene can be retrieved from the source metagenome dataset using targeted (manual) binning. This last step will not always be successful, because less sequencing depth is required to assemble a single short contig consisting of one gene, rather than assemble longer contigs and confidently assign them to a draft genome. However, precisely because more sequencing depth is required to assemble and bin a draft genome than a single gene, marker gene mining might yield information from datasets where genome binning is not feasible (e.g. *Lüke et al., 2016*).

We have previously used this approach to assess the presence of nitrogen cycle genes in datasets from the Arabian Sea oxygen minimum zone (*Lüke et al., 2016*) and to assess the environmental distribution of organisms capable of complete ammonium oxidation (*Van Kessel et al., 2015*). Here we present a more systematic use of marker gene mining to assess the diversity of the *mcrA* gene, encoding the alpha subunit of the methyl-coenzyme M reductase. This enzyme is essential for (reverse) methanogenesis, where it catalyzes the final reduction and release of the methyl group on coenzyme M to methane, or the initial oxidation of methane (*Nagle & Wolfe, 1983*; *Scheller et al., 2010*). Moreover, the *mcrA* gene has recently been discovered in several unexpected clades of *Archaea*, indicating that methane metabolism is more widespread in the domain than previously thought (*Mondav et al., 2014*; *Evans et al., 2015*; *Vanwonterghem et al., 2016*; *Sorokin et al., 2017*). Notably, *Syntrophoarchaeum*, an archaeon in a syntrophic anaerobic butane-oxidizing enrichment culture, contained multiple copies of highly divergent *mcrA* genes thought to be involved in the activation of higher alkanes, but not methane (*Laso-Pérez et al., 2016*).

We have screened the environmental metagenomic data available in the NCBI sequencing read archive (*Kodama et al., 2012*) and MG-RAST (*Meyer et al., 2008*) for reads matching the mcrA gene. We subsequently used the obtained data to assess the diversity and environmental distribution of the *Methanomassiliicoccales* order (*Paul et al., 2012*). Finally, we reconstruct and analyze a draft genome of an organism belonging to the 'Lake Pavin cluster' an uncultivated lineage within this group (*Borrel et al., 2013b*).

## METHODS

### Construction of the mcrA database from PFam and NCBI-nr

To construct a mcrA protein sequence database representative of the known global diversity, we first obtained the amino acid sequences in the Pfam (version 29.0; *Finn et*

*al., 2013*) families MCR_alpha_N (PF02745) and MCR_alpha (PF02249), representing the N-terminal and C-terminal parts of the mcrA protein. We assessed the completeness of the Pfam dataset by downloading the NCBI-nr protein reference database in fasta format (ftp://ftp.ncbi.nlm.nih.gov/blast/db/FASTA/) and using it as query for a DIAMOND (*Buchfink, Xie & Huson, 2014*) search against the Pfam dataset. Subsequently, we calculated the BLAST score ratio (BSR) (*Rasko, Myers & Ravel, 2005*) between the score against the Pfam dataset and the maximum possible score (a self-hit) of the 16,260 sequences that had a DIAMOND hit against the Pfam dataset (Fig. S1). This allowed us to identify real mcrA sequences not covered in the Pfam, while eliminating false positive hits. Using this method, we identified that mcrA sequences from the clades *Bathyarchaeota*, *Methanofastidiosales* (WSA2), ANME-1, *Methermicoccus* and *Methanoperedens* (ANME-2d) were not represented in the Pfam at the time of database construction (June 2016) (Fig. S1). All 203 full-length mcrA sequences in the NCBI-nr were added to the mcrA dataset, which was subsequently clustered at 90% identity using UCLUST (*Edgar, 2010*), resulting in a mcrA reference database containing 69 non-redundant full-length sequences representing the full diversity of mcrA sequences included in the NCBI-nr (June 2016).

## mcrA read data acquisition from SRA and MG-RAST

To obtain a list of metagenome datasets of potential interest, metadata was downloaded for all runs in the sequencing read archive (SRA) (https://www.ncbi.nlm.nih.gov/sra) corresponding to the query "metagenomic AND WGS NOT human NOT gut NOT oral" (gut and oral were excluded because of the high number of datasets from these environments) on June 16th 2016. Additionally, metadata for all runs in MG-RAST (https://metagenomics.anl.gov/) was downloaded, and all datasets labeled 'WGS' (whole genome shotgun) were selected. This resulted in a list of 10,613 SRA run accession numbers and 7,852 MG-RAST identifiers, representing over 60 Terabases of sequencing data (Fig. 1, Files S1 and S2). As storing this amount of sequence data was not feasible, the accession lists were used as input for the 'sra_trawler.sh' and 'mgrast_trawler.sh' shell scripts, respectively. Briefly, these scripts download an accession number on the list, then use the downloaded dataset as query in a DIAMOND (*Buchfink, Xie & Huson, 2014*) search against a database of interest (in this case the mcrA protein database described above). Then, hits are written to a new file in fasta format using the script 'blast_based_read_lookup.pl', the dataset is discarded, and the process is repeated for the next dataset. Scripts are available at (https://github.com/dspeth/bioinfo_scripts/tree/master/metagenome_screening). To speed up the process, the sra-trawler.sh script uses the sra-toolkit (https://www.ncbi.nlm.nih.gov/sra/docs/toolkitsoft/) to split paired-end files and only analyzes the forward reads of paired-end datasets. Using this approach, it took approximately 4 months to process the 18,465 selected datasets using 16 cores on a server. 2,083,349 reads from 6,105 datasets matched the mcrA database and were combined and used as the query in a DIAMOND search against the NCBI-nr database to calculate the BSR between the hit against the mcrA database and the NCBI-nr (Fig. 1). This results in a BSR between the score against the database of interest and the score against an outgroup, rather than a BSR between the score against the database of interest and a self-hit described above for mcrA database

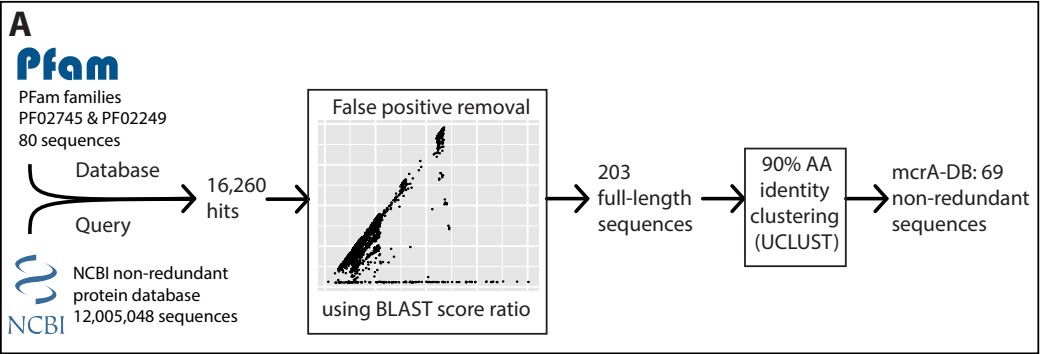

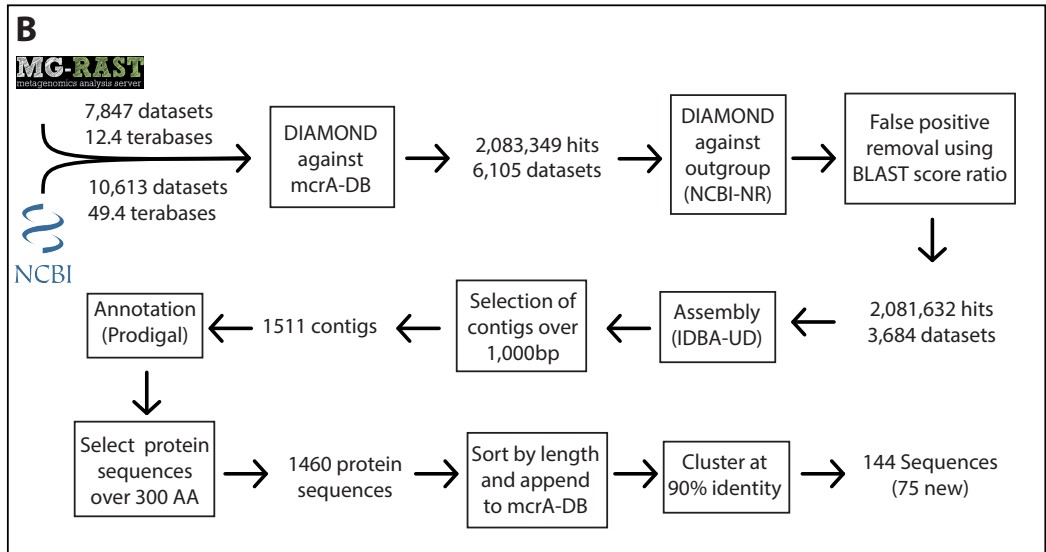

**Figure 1** **Overview of the marker gene mining workflow.** (A) Construction of the mcrA reference database using the two mcrA Pfam families and the NCBI non-redundant protein database. (B) Screening metagenomes in the sequencing read archive and MG-RAST for the presence of *mcrA* and reconstructing *mcrA* gene sequences.

construction. Applying BSR in this manner allows for detection of real mcrA sequences with low identity to the database while still removing false positives (*Lüke et al., 2016*). The resulting 2,081,632 post-BSR hits originated from 3,684 datasets (Fig. 1). Dataset SRR398144, an mcrA amplicon sequencing effort (*Denonfoux et al., 2013*), accounted for nearly 10% of these hits (197,371). As the aim of our work was to reconstruct near full-length mcrA sequences, SRR398144 was excluded and the remaining steps were done with 1,884,261 reads.

## Assessing global mcrA protein diversity

The true-positive reads from 1,080 datasets containing over 20 hits, enough for approximately two fold gene coverage in a 150 bp dataset, were assembled separately using IDBA-UD (*Peng et al., 2012*), resulting in 1,511 sequences >1,000 bp. Prodigal (*Hyatt et al., 2010*; v2.6.2) was used in single mode and without the –c flag, allowing open

ends, for open reading frame prediction on the assembled contigs and mcrA protein sequences over 300 amino acids were selected. The resulting 1,460 sequences were sorted by length and added to the mcrA database. Additionally, the eight recently published *Syntrophoarchaeum* mcrA sequences (*Laso-Pérez et al., 2016*), not available at the time of initial database construction and mining, were added to the mcrA database, and the new set was clustered using UCLUST at 90% identity (*Edgar, 2010*). The resulting 150 sequences were aligned using MUSCLE (*Edgar, 2004*) and a maximum likelihood phylogeny was calculated using RAxML (*Stamatakis, 2014*), with the LG4X substitution model (*Le, Dang & Gascuel, 2012*) and 500 bootstrap replicates. The phylogeny was visualized using iTOL (*Letunic & Bork, 2016*; Fig. 2). To estimate previous detection of the mcrA sequences, all 15,888 hits post-BSR selection in the NCBI-nr that were too short to be included in the database (PCR products) were used as DIAMOND query against the 150 non-redundant mcrA sequences (Fig. S2A). Additionally, all 1,884,261 metagenome hits were used as DIAMOND query against the 149 non-redundant mcrA sequences (Fig. S2B). Relative counts of all 13,007 NCBI-nr hits and 1,487,226 metagenomic reads over 90% identity (as these would be clustered with the sequence) were visualized using iTOL. Reconstructed gene sequences are included as Files S3 and S4, containing the nucleotide and amino acid sequences respectively.

## Environmental distribution of the *Methanomassiliicoccales*

The assembled mcrA sequences belonging to the *Methanomassiliicoccales* clade were obtained using BSR between the best hit against the five *Methanomassiliicoccales* mcrA sequences present in our mcrA database, before addition of the newly assembled sequences (Fig. 2), and the best hit against our entire mcrA database. Sequences with a BSR over 0.75 were assigned to the *Methanomassiliicoccales* (Fig. S2). The resulting 116 sequences, combined with the five reference sequences discussed above and a *Methanofastidiosales* sequence (KYC45731.1, as outgroup), were aligned using MUSCLE (*Edgar, 2004*) and a phylogeny was calculated using RaxML (*Stamatakis, 2014*) with the LG4X substitution model (*Le, Dang & Gascuel, 2012*) and 500 bootstrap replicates. The resulting phylogeny was visualized using iTOL (*Letunic & Bork, 2016*). The source environment of each sequence was assigned manually, using the NCBI-SRA or MG-RAST sample record.

## Genome reconstruction of a Lake Pavin cluster *Methanomassiliicoccales*

Reads from dataset SRR636597 (*Tan et al., 2015*) were assembled using Megahit (*Li et al., 2015*), (v.1.0.3, with presets meta-large) resulting in 125,408 contigs longer than 1,000 bp. Reads from datasets SRR636597, SRR636559, and SRR636569, which contained highly similar mcrA sequences (Fig. 3), were mapped to the assembled contigs using BBmap (http://jgi.doe.gov/data-and-tools/bbtools/bb-tools-user-guide/) using a cutoff of 95% identity over 80% of the read length. Kmer frequency of the contigs was calculated using the script calc.kmerfreq.pl by Mads Albertsen (https://github.com/MadsAlbertsen/multi-metagenome/tree/master/R.data.generation). The contig fasta file was converted to tab-delimited form and length and GC content were calculated using fasta_to_gc_length_tab.pl

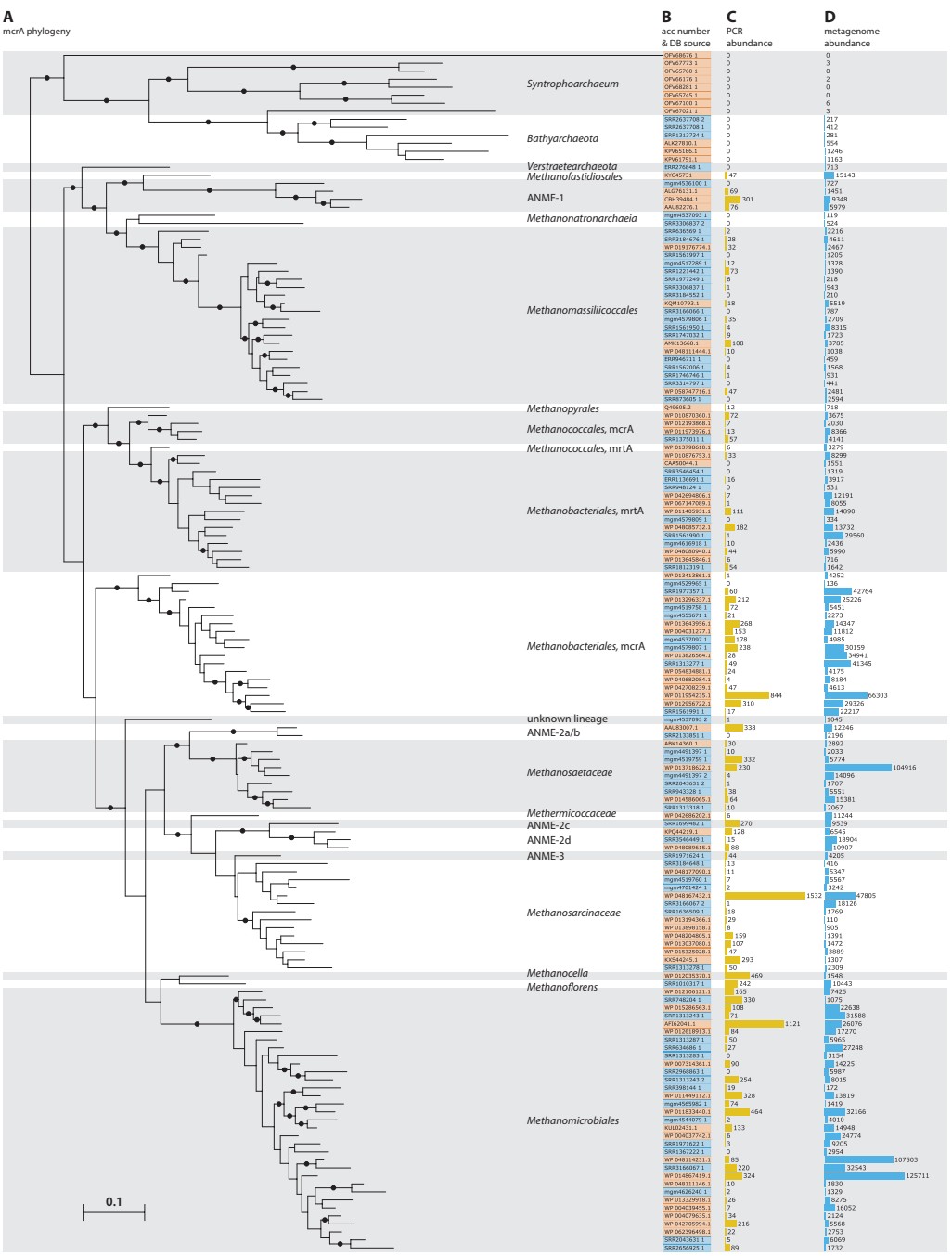

**Figure 2 Phylogeny and environmental detection of recovered mcrA sequences.** (A) Maximum likelihood phylogeny of the translated mcrA sequences. Background shading is used to delineate major clades. Bootstrap values over 70% are indicated by black circles. The two copies of *mcrA* in *Methanococcales* and *Methanobacteriales* are indicated with *mcrA* & *mrtA*. (B) Protein accession number or dataset accession number of the sequences in the phylogeny. Sequences obtained from the NCBI-nr are highlighted in orange, sequences assembled in this study from SRA and MG-RAST datasets are highlighted in blue. (C) Number of sequences present in the NCBI-nr with over 90% amino acid identity to the sequences in the phylogeny. This includes mostly gene fragments amplified using PCR. (D) Number of sequencing reads, after BLAST Score Ratio (BSR) filtering, from metagenomes in the SRA and MG-RAST with over 90% amino acid identity to the sequences in the phylogeny.

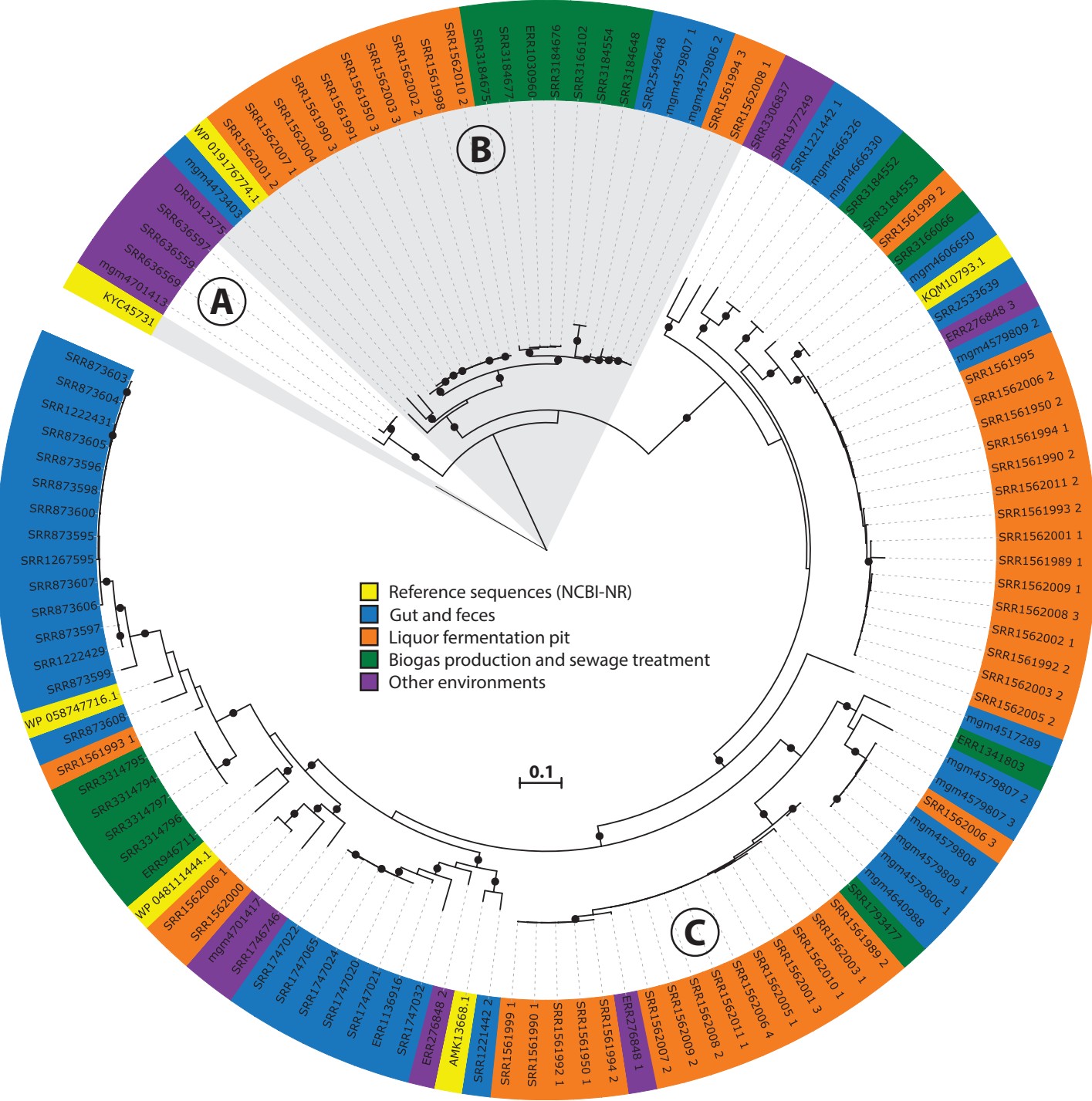

**Figure 3** **Phylogeny and environmental distribution of the mcrA sequences within the *Methanomassiliicoccales* order.** Maximum likelihood phylogeny of the translated *mcrA* sequences belonging to the *Methanomassiliicoccales* order retrieved in this study, and the five *Methanomassiliicoccales mcrA* sequences present in the NCBI-nr, after dereplication at 90% identity, at the time of database construction. Bootstrap values over 70% are indicated by black circles. Leaf labels are accession numbers of the protein sequence (in the case of the five reference sequences) or source dataset. Coloring of leaf labels indicates source environment. Shading indicates the three clusters discussed in the text: (A) Lake Pavin cluster, (B) Environmental cluster, (C) Gastrointestinal tract (GIT) cluster.

(https://github.com/dspeth/bioinfo_scripts/tree/master/metagenome_screening). The contig containing the mcrA sequence of interest was identified using DIAMOND (*Buchfink, Xie & Huson, 2014*) and used to guide the manual binning of the cluster representing the draft genome containing this gene, using R, as previously described (*Speth et al., 2016*). The resulting 1.6 Mbp draft genome was further refined by 10 cycles of mapping with BBmap and assembly with SPAdes (*Bankevich et al., 2012*) using the iterative_denovo_spades.sh shell script (https://github.com/dspeth/bioinfo_scripts/tree/master/metagenome_screening). A final manual refining was performed using anvi'o v4, using clustering based on composition of the contigs as well as their coverage in datasets SRR636597, SRR636559, and SRR636569 to remove contaminating contigs from the bin (*Eren et al., 2015*). The final 1.55 Mb draft genome on 58 contigs was quality checked using checkM (v1.0.6) with the default settings of the lineage_wf command (*Parks et al., 2015*), annotated using Prokka (v1.12), with flags –compliant, –metagenome, and –kingdom Archaea (*Seemann, 2014*), analyzed using Artemis (*Carver et al., 2012*), compared to the other available *Methanomassiliicoccales* genomes using ProteinOrtho with default settings (*Lechner et al., 2011*; v5.12), and the annotation was manually curated by verifying the presence/absence of the complexes discussed in the text. As NCBI does not accept third party annotation without experimental validation (see: https://www.ncbi.nlm.nih.gov/genbank/tpa/), the assembled annotated draft genome is included as Files S5.

## RESULTS AND DISCUSSION

### McrA gene diversity across all sampled datasets

To leverage the available metagenomic sequencing data for a diversity analysis of functional marker genes, we established a workflow based on automated sequential downloading and processing of the public data in the Sequencing Read Archive (SRA) and the Metagenomics RAST (MG-RAST) repositories (Fig. 1). We applied this workflow to the *mcrA* gene, a marker for the production and anaerobic oxidation of methane, because of the environmental relevance of these processes (*Knittel & Boetius, 2009*). The recently discovered mcrA sequences of the *Verstraetearchaeota*, *Syntrophoarchaea*, *Methanonatronarchaeia*, and *Methanoflorens* were not present in the NCBI-nr as of June 2016, and thus not included in our reference database. Of these, our analysis does retrieve mcrA sequences associated with *Verstraetearcheota*, *Methanonatronarchaeia*, and *Methanoflorens* (Fig. 2), but was not sensitive enough to retrieve sequences related to the highly divergent *Syntrophoarchaea*, simultaneously illustrating both the power to detect novel diversity, and the limit of screening of metagenomic reads based on sequence identity. The HMM based search strategy implemented in GraftM (*Boyd, Woodcroft & Tyson, 2018*), offers a complementary strategy for mining unassembled reads, further increasing the potential for discovery of novel diversity.

Besides the independent recovery of the *Verstraetearcheota*, *Methanonatronarchaeia*, and *Methanoflorens* from public databases, two sequences deeply branching within the *Euryarchaeota* were retrieved. Both sequences were assembled from dataset mgm4537093.3, originating from a marine sediment sample at the oil seeps of the coast of Santa

Barbara, California. One of these sequences is basal to the *Methanomassiliicoccales*, and distantly related to *Methanonatronarchaeia*, and the other sequence is basal to the *Methanosarcinales/Methanocellales/Methanomicrobiales* cluster (Fig. 2). However, the organisms containing these sequences were not present in sufficient abundance in their respective samples to extract a draft genome from the metagenome, leaving the taxonomic association of these divergent sequences unclear. Furthermore, several sequences associated with anaerobic methanotrophic (ANME) archaea were retrieved (Fig. 2), including a sequence from a South African gold mine (*Lau et al., 2014*) related to ANME-1, and the first full length ANME-3 mcrA sequence from a dataset obtained from the deep-sea Haakon Mosby mud volcano, the site where this group was originally discovered (*Niemann et al., 2006*; *Lösekann et al., 2007*). Our analysis also substantially expands the known diversity of mcrA sequences of methanogenic clades within the *Euryarchaeota* (Fig. 2) and allows for an estimate of environmental abundance of these clades. Such an abundance estimate is likely affected by sampling bias, for example by an overrepresentation of biogas fermentation reactors. There are clear differences comparing an abundance estimate based on PCR products in the NR (Fig. S1) with an estimate by metagenomic reads (Fig. 2) possibly due to primer bias, or a consequence of dereplication of the PCR products in the NR, either by NCBI (removing redundant sequences) or prior to submission.

Aligning the BSR-filtered reads from our analysis to the mcrA database amended with the newly assembled sequences shows higher average sequence identity of the aligned sequences after addition of the new sequences. Before our analysis, 555,598 reads (29.5% of BSR-filtered reads) had lower than 90% identity to a sequence in our reference database, whereas after addition of the newly retrieved sequences that number dropped to 313,904 reads below 90% identity (16.6% of BSR-filtered reads; Fig. 2; Fig. S3). For the PCR amplicons present in the NCBI-nr (Fig. S1) these numbers were comparable, with 4628 (29.1%) sequences below 90% identity to any sequence in the database before including the newly retrieved sequences, and 2841 (17.9%) below 90% identity afterwards. Although this is an improvement, the high number of reads still unassigned (>90% sequence identity to any database sequence) does indicate there are yet more divergent mcrA variants to be discovered (Fig. S3). This highlights the potential for ongoing exploration of metagenomic sequencing data as it becomes available, using our sequence-identity based analysis, and an HMM based approach (*Boyd, Woodcroft & Tyson, 2018*). The highest number of novel sequences retrieved in our survey of the SRA and MG-RAST was associated with the *Methanomassiliicoccales*, bringing the known diversity within this recently discovered order on par with that of more intensively studied groups.

## Environmental distribution of the *Methanomassiliicoccales*

To further investigate the diversity and environmental distribution of the *Methanomassiliicoccales*, we retrieved all 116 mcrA sequences over 300 amino acids belonging to this clade from our analysis and assessed their environmental origin as documented in the SRA or MG-RAST metadata record. This analysis confirmed the presence of three major clades ('GIT', 'Environmental' and 'Lake Pavin'), but unlike the study by Söllinger et al. we do not observe clear clustering by environment (Fig. 3). Even though our original dataset

selection was biased against gut samples by the SRA query used, several sequences of fecal origin are represented in both the environmental and GIT clades (Fig. 3). Conversely, both the 'Environmental' and 'GIT' clades contain many sequences originating from the same environment; a single study that characterizes the microbial diversity in open fermentation pits for liquor production (*Guo et al., 2014*). None of the assembled sequences originated from marine sediments, implying that the *Methanomassiliicoccales* are not abundant in these systems, and that other archaeal clades (possibly) associated with the *Thermoplasmatales* are responsible for the detected BDGT lipids (*Becker et al., 2016*).

Four sequences, of which three were nearly identical, belonging to the 'Lake Pavin' clade were retrieved in our analysis (Fig. 3). As this group was previously only detected by environmental PCR, and genomic data is lacking, we focused on the three near-identical sequences for more in-depth analysis. Using the mcrA sequences as a guide, we extracted a draft genome of a representative of this clade from dataset SRR636597, originating from an oil mining tailing pond (*Tan et al., 2015*). This draft genome is referred to as MALP (MAssiliicoccales Lake Pavin) for the remainder of the manuscript.

## Genomic analysis of MALP, a representative of 'Lake Pavin' clade *Methanomassiliicoccales*

The MALP draft genome assembled and binned from dataset SRR636597 consists of 1.55 megabases on 58 contigs, and is over 92% complete with 0% contamination, as assessed using checkM (*Parks et al., 2015*). A recent assembly and binning effort of 1,550 SRA datasets recovered a highly similar MAG, likely representing the same microbial population, designated UBA248 (*Parks et al., 2017*). At an estimated 89% completeness, 1.6% contamination, and on 138 contigs UBA248 is somewhat more fragmented and slightly lower quality. Furthermore, Parks et al. did not perform any analysis on this specific MAG. In addition, Parks et al. also assembled a related MAG, designated UBA472, from a different site. Contig alignment using Mauve (*Rissman et al., 2009*) and comparison of gene content using anvi'o (*Eren et al., 2015*) supports that MALP and UBA248 represent the same population, whereas UBA472 likely represents a closely related organism (Fig. S4).

The size of the MALP genome falls within the range of previously obtained *Methanomassiliicoccales* genomes (1.4–2.6 Mbp; *Borrel et al., 2014*; *Lang et al., 2015*). As in the other *Methanomassiliicoccales*, the ribosomal rRNA genes in MALP are not organized in an operon, and it encodes two copies of the 5S rRNA gene. Our mcrA analysis and BLAST searches using the MALP 16S rRNA gene indicate that organisms closely related to MALP (>98% 16S rRNA gene identity) are previously found in (contaminated) sediments and wastewater treatment systems, but not in fecal samples.

MALP encodes all genes required for hydrogen dependent reduction of methanol to methane as proposed in the other *Methanomassiliicoccales* (Fig. 4A). These include the methanol:CoM methyltransferases (*mtaABC*) for formation of methyl-CoM (*Sauer & Thauer, 1999*), methyl-CoM reductase (*mcrABG*) to release methane and form the CoM-CoB heterodisulfide (*Ermler et al., 1997*), soluble heterodisulfide reductase (*hdrABC*) and [NiFe]-hydrogenase (*mvhAGD*) to reduce ferredoxin and heterodisulfide coupled to hydrogen oxidation (*Thauer et al., 2008*; *Wagner et al., 2017*), and the Fpo-like complex

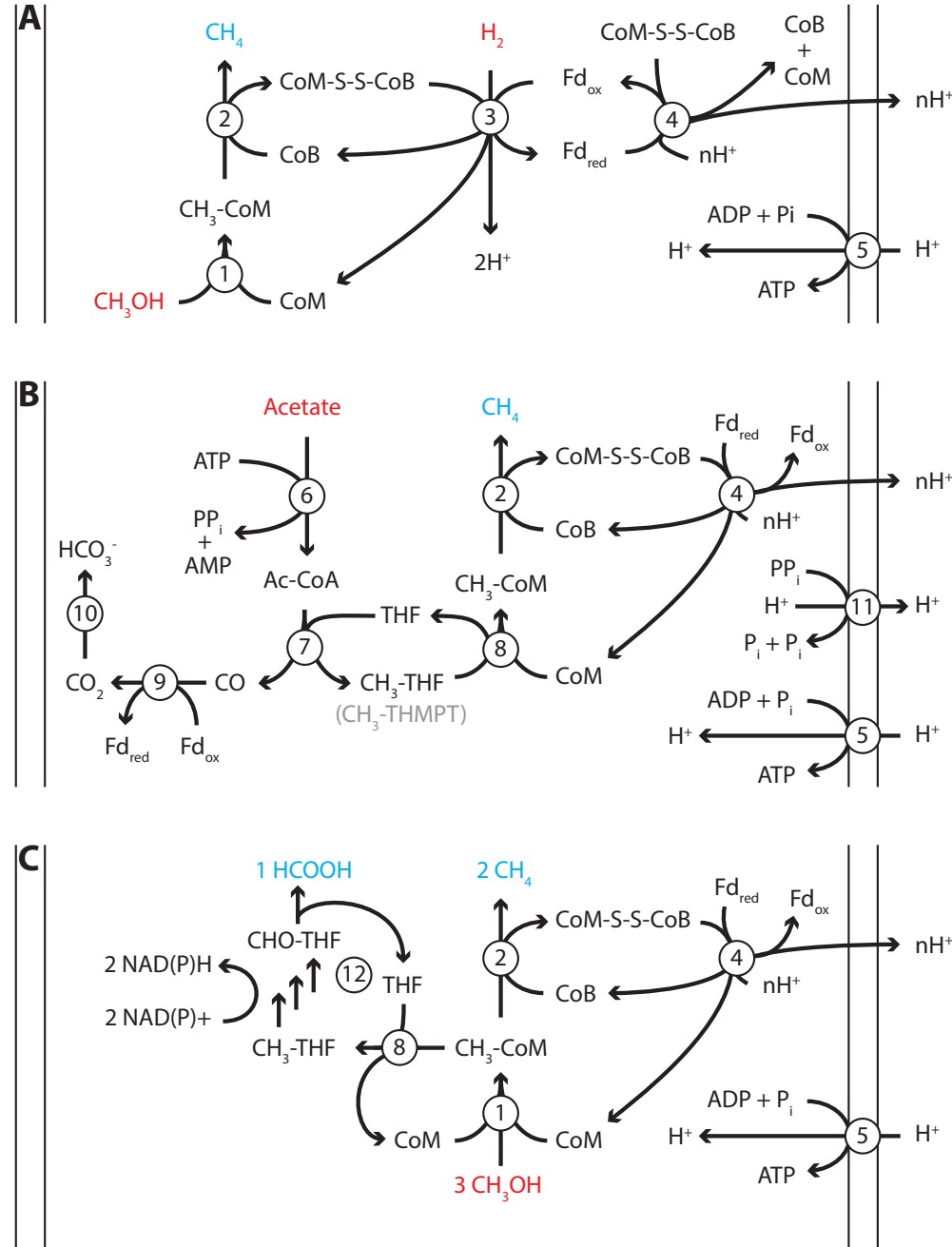

**Figure 4** **Proposed energy metabolism of the *Methanomassiliicoccales* Lake Pavin (MALP) cluster genome.** Proposed energy metabolism in the MALP genome. (A) Hydrogen dependent reduction of methanol to methane. (B) Acetate disproportionation to methane and carbon dioxide. (C) Methanol disproportionation to methane and formate. Substrates of energy metabolism are indicated in red, products in blue. Enzymes are indicated by numbered circles: (1) Methanol:coM methyltransferase; (2) methyl-coenzyme M reductase; (3) [NiFe]-hydrogenase/heterodisulfide reductase; (4) Fpo-like complex/heterodisulfide reductase; (5) ATP synthase; (6) Acetyl-CoA synthetase; (7) Acetyl-CoA synthase; (8) N5-tetrahydrofolate:Coenzyme M methyltransferase; (9) CO-dehydrogenase; (10) carbonic anhydrase; (11) energy conserving pyrophosphatase; (12) N5-methyltetrahydrofolate oxidation pathway. CoM, Coenzyme M; CoB, Coenzyme B; Fd, ferredoxin; THF, tetrahydrofolate; THMPT, tetrahydromethanopterin.

(*fpoABCDHIJKLMN*) proposed to oxidize ferredoxin and establish a proton gradient (*Welte & Deppenmeier, 2011*), potentially coupled to heterodisulfide reduction using *hdrD* (Fig. 4) (*Lang et al., 2015*). Like "*Candidatus* Methanomethylophilus alvus", MALP does not encode an energy conserving hydrogenase to couple ferredoxin oxidation to hydrogen formation and the buildup of a proton gradient (*Borrel et al., 2014*).

From both physiological studies on the *Methanomassiliicoccus luminyensis* culture and the "*Candidatus* Methanomethylophilus alvus" enrichment, as well as comparative genomics, it has become clear that previously enriched *Methanomassiliicoccales* are also capable of growth on methylamines (*Borrel et al., 2014*; *Lang et al., 2015*). Metabolizing methylamines requires pyrrolysine-containing methyl transferases (*mtmBC/mtbBC/mttBC*), which are present in the genomes of the previously sequenced *Methanomassiliicoccales*. In contrast, the reconstructed MALP genome does not encode the pyrrolysine-containing methyltransferases, the operon for pyrrolysine biosynthesis, the pyrrolysine tRNA synthetase, or the pyrrolysine tRNA. Considering that MALP is a representative of the most basal cluster of *Methanomassiliicoccales* (Fig. 3), the absence of pyrrolysine usage from the genome suggests the ability to generate methane from methylated amines was acquired recently within the *Methanomassiliicoccales* order. In agreement with this, a previous comparative genomics study found that the number of pyrrolysine containing genes in other sequenced *Methanomassiliicoccales* varied between three (*Methanomassiliicoccus luminyensis*) and 19 ('*Candidatus* Methanomethylophilus alvus') (*Borrel et al., 2014*).

Another unexpected feature of the MALP genome was the presence of the two catalytic subunits of N5-methyltetrahydromethanopterin:CoM methyltransferase (*mtrAH*) (*Wagner, Ermler & Shima, 2016*). In $CO_2$-reducing methanogens methyltetrahydromethanopterin:CoM methyltransferase is an eight-subunit membrane-associated complex that catalyzes the transfer of a methyl group from tetrahydromethanopterin (THMPT) to coenzyme M, coupled to translocation of a sodium ion. However, only the catalytic subunits (*mtrAH*), and none of the membrane associated subunits (*mtrB-G*), are present in the MALP genome. This is surprising, as there is no known role for THMPT in *Methanomassiliicoccales*, including MALP. In contrast, MALP does encode the complete C1-tetrahydrofolate (THF) pathway. The *mtrA* subunit, which donates the methyl group to Coenzyme M is conserved, but contains an unusual C-terminal extension. On the other hand, the *mtrH* subunit that is responsible for the transfer of the methyl group from THMPT is divergent from the *mtrH* of $CO_2$-reducing methanogens. Considering the likely absence of THMPT from MALP, and the divergent *mtrH* subunit, we propose this *mtrAH* may be a N5-methyltetrahydrofolate:CoM methyltransferase instead (Fig. 4B).

In addition to the *mtrAH* genes, MALP encodes acetyl-CoA synthetase for the formation of acetyl-CoA from acetate (*Jetten, Stams & Zehnder, 1989*), and acetyl-CoA synthase/CO dehydrogenase for the disproportionation of acetyl-CoA into methyltetrahydromethanopterin or methyl-tetrahydrofolate and CO, and ferredoxin dependent oxidation of CO to $CO_2$ (Fig. 4B) (*Ferry, 1992*). This gene set, combined with the presence of the Fpo-like complex, in theory provides MALP with a complete pathway for aceticlastic methanogenesis, similar to the pathway observed in *Methanosaeta thermophila* (Fig. 4B)

(*Welte & Deppenmeier, 2014*). Even though genes required for aceticlastic methanogenesis were detected, there are a number of differences between the pathway in the obligate aceticlastic methanogen *Methanosaeta thermophila*, and the hypothetical pathway in MALP that make it doubtful that MALP has the capability to produce methane from acetate.

Notably, MALP lacks the membrane complexes thought to conserve energy for ATP production in *Methanosaeta thermophila* (*Welte & Deppenmeier, 2014*); the membrane subunits of the *mtr* complex (*mtrBCDEFG*) and the membrane-bound heterodisulfide reductase (*HdrDE*) are absent. More specifically, the two ATP equivalents expended in the conversion of acetate to acetyl-CoA using acetyl-CoA synthetase require a minimum translocation of seven protons/sodium atoms to regenerate the ATP (at three charge translocations/ATP) and build up a potential of 1 net proton/sodium atom per molecule of acetate. *M. thermophila* might achieve this by translocating 2 $Na^+$ atoms using the *mtr* complex, two protons using the membrane-bound heterodisulfide reductase (*HdrDE*), and three protons using the Fpo-like complex (*Welte & Deppenmeier, 2014*). Compared to *M. thermophila*, MALP lacks the sodium translocating subunits of *mtr* (*mtrBCDEFG*) and the integral membrane subunit of the membrane-bound heterodisulfide reductase (*HdrE*). However, MALP does encode an energy conserving pyrophosphatase, and the *hdrD* subunit of membrane bound heterodisulfide reductase has been proposed to interact with the Fpo-like complex, potentially raising the number of protons translocated by the Fpo complex to four (*Lang et al., 2015*). In addition, MALP encodes an "energy-conserving hydrogenase related" (*ehr*) complex, first observed in *Geobacter sulfurreducens* (*Coppi, 2005*). The function of this complex is unknown, but it harbors several proton-pumping subunits (*Marreiros et al., 2013*) and could be involved in energy conservation in MALP. Although these complexes could account for sufficient charge translocation, there is not enough biochemical evidence to confidently predict MALP has the ability to produce methane from acetate.

In addition to a pathway for aceticlastic methanogenesis, the presence of the *mtrAH* genes, combined with the pathway for N5-methyltetrahydrofolate oxidation to formate, could enable MALP to grow using an unconventional type of methanol disproportionation shown in Eq. (1) and Fig. 4C.

$$3\,CH_3OH \rightarrow HCOO^- + H^+ + 2CH_4 + H_2O\,(\Delta G^{0'} = -203.4\ kJ/mol) \tag{1}$$

$$4\,CH_3OH \rightarrow CO_2 + 3CH_4 + 2H_2O\,(\Delta G^{0'} = -319.4\ kJ/mol). \tag{2}$$

Although thermodynamically feasible (based on $\Delta G_f^0$ from *Thauer, Jungermann & Decker, 1977*), there are various caveats to this proposed metabolism. First, it is less energetically favorable than methanol disproportionation to methane and $CO_2$ Eq. (2), suggesting that the metabolism would not be competitive in the environment. However, when expressed per mol substrate (methanol) the energy difference drops to 67.8 kJ/mol methanol versus 79.85 kJ/mol methanol for disproportionation to formate/methane and $CO_2$/methane respectively. This relatively small difference could be overcome by efficient formate removal by other organisms in the environment.

Another caveat is that the endergonic methyl transfer from Coenzyme M to THMPT in methanol disproportionating methanogens is thought to be driven by dissipation

of a sodium gradient. As *mtrAH* in MALP is likely not membrane associated, this mechanism seems unlikely. However, methyl transfer from CoM to tetrahydrofolate (THF) is likely less endergonic than transfer to THMPT (*Chistoserdova et al., 1998*; *Maden, 2000*) and might proceed without being driven by a sodium gradient. The remainder of the C1-THF pathway is reversible, albeit less favorable than the C1-THMPT pathway (*Maden, 2000*). The entire C1-THF pathway is also present in *Methanomassiliicoccus lumiyensis* and *Methanomassiliicoccus intestinalis*, while the other *Methanomassiliicoccales* lack only the gene for conversion between N5-methyltetrahydrofolate and 5–10 methylenetetrahydrofolate (*metF*). This pathway is also proposed to be used in the oxidative direction, to supply intermediates for purine biosynthesis (*Lang et al., 2015*). None of the *Methanomassiliicoccales*, including MALP, encode formate dehydrogenase (*Lang et al., 2015*). Oxidation of the methyl group would thus stop at formate, generating four electrons and resulting in the stoichiometry shown in Eq. (1). A final caveat with this proposed pathway is the conversion between NAD(P)H generated in the oxidation of N5-methyltetrahydrofolate to formate, and the ferredoxin that is oxidized at the Fpo-like complex. MALP does not encode a homolog of Ferredoxin:NADP reductase, thus at present the candidate for this reaction is unknown.

In summary, the MALP genome, belonging to the 'Lake Pavin' clade of the *Methanomassiliicoccales*, indicates that the MALP organism is a hydrogenotrophic methyl-reducing methanogen, capable of growth on methanol. Unlike the other members of the *Methanomassiliicoccales*, MALP does not encode the genes required for growth on other methylated compounds, such as methylamines or methylsulfides. However, MALP does encode an unusual *mtrAH* complex, which might allow for aceticlastic methanogenesis, as well as methanol disproportionation. However, as outlined above, the latter two predicted metabolic capabilities are highly uncertain without further physiological and biochemical data. Therefore, obtaining a cultured representative of this clade would greatly aid in testing these hypotheses.

## CONCLUSIONS

Metagenomic marker gene mining is a complementary approach to genome resolved metagenomics, and can be used to assess phylogenetic diversity and environmental distribution of a microbial process. Due to the size (and continued rapid growth) of public sequence databases our implementation is slow. However, this approach is broadly applicable as it requires minimal computational power due to the small database size, and minimal storage (in contrast to locally storing a version of the public databases) because datasets are processed sequentially and then deleted. Using this marker gene mining approach, we recovered novel *mcrA* gene diversity, and identified promising targets for more in-depth analysis leading to better understanding of the habitat distribution and metabolic versatility of the environmental *Methanomassiliicoccales*. Marker gene mining can be used to query the large amount of data from the many publicly available sequencing projects for specific questions, and potentially lead to discoveries outside the scope of the original studies. The rapidly increased throughput, and reduced cost, of next generation

sequencing ensures that much more data will become available in years to come, and complementary strategies to analyze this sequencing data effectively will be increasingly important going forward.

# ACKNOWLEDGEMENTS

We thank Woody Fischer and Connor Skennerton for helpful discussion and Grayson Chadwick for critically reading the manuscript.

## Funding
This manuscript is based upon work supported by the US Department of Energy, Office of Science, Office of Biological and Environmental Research under award number DE-SC0016469 to Victoria J. Orphan. In addition, Daan R. Speth was supported by NWO Rubicon 019.153LW.039. The funders had no role in study design, data collection and analysis, decision to publish, or preparation of the manuscript.

## Grant Disclosures
The following grant information was disclosed by the authors:
US Department of Energy, Office of Science, Office of Biological and Environmental Research: DE-SC0016469.
NWO Rubicon: 019.153LW.039.

## Competing Interests
The authors declare there are no competing interests.

## Author Contributions
- Daan R. Speth conceived and designed the experiments, performed the experiments, analyzed the data, contributed reagents/materials/analysis tools, prepared figures and/or tables, authored or reviewed drafts of the paper, approved the final draft.
- Victoria J. Orphan conceived and designed the experiments, authored or reviewed drafts of the paper, approved the final draft.

## Data Availability
GitHub: https://github.com/dspeth/bioinfo_scripts/tree/master/metagenome_screening.

## Supplemental Information
Supplemental information for this article can be found online at http://dx.doi.org/10.7717/peerj.5614#supplemental-information.

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
