# Peer review of "Metabolic marker gene mining provides insight in global mcrA diversity and, coupled with targeted genome reconstruction, sheds further light on metabolic potential of the Methanomassiliicoccales"

_PeerJ, doi:10.7717/peerj.5614_

## Round 0.1 · original submission · Minor Revisions

Most of the reviewers' comments assess the organization of the paper, the referencing, and some terminology, although Reviewer 2 also has some comments on the analytical methods.

Two reviewers suggest that the paper would be more readable with a combined "Results and Discussion" section. This is not the standard format for PeerJ, but it is allowed in principle. You may consider this suggestion if you agree that it will improve the paper, but may also keep the journal-standard format with separate sections.

-Peter

·

Basic reporting

This manuscript is very clearly formulated, logically structured, well-referenced, and uses figures of good quality. It is therefore easy to follow.

Experimental design

The experimental design is excellent and outlined in a detailed manner that will be helpful to researchers performing similar metagenomic-marker gene based analyses in the future. The importance and advantage of the proposed approach over computationally even more intensive screening methods are clear.

Validity of the findings

The findings are valid, and the authors do not overstate the importance of their findings.

Additional comments

Title:
The term "metabolic versatility" in the title suggests to this reader the ability to catabolize a wide spectrum of substrates or perform active metabolism under a wide range of environmental conditions, neither of which were reflected in the findings. I suggest replacing "metabolic versatility" witha more neutral term, e.g. "metabolism".

Abstract:
The abstract is almost exclusively focused on the method and its potential applications. Since the entire Discussion is devoted to the analysis of metabolic capability and the title also points out the important role of metabolic reconstruction in this manuscript, I suggest adding 2-3 sentences on inferred/potential metabolic traits here. This should include a cautious mentioning of the presence of genes that could enable aceticlastic methanogenesis, since this is an important finding even though physiological studies are necessary to demonstrate this form of catabolism in the actual organism. In turn, other parts in the abstract, e.g. the introduction of the method, details on the methodology, could be made more concise or streamlined to only feature information that is essential for a general understanding of the approach used.

L. 12 in Abstract: unclear what is meant with "understudied". Please use a more precise term.

Introduction:
L. 65-66: this sentence is too strong. mcrA genes are excellently suited for the reconstruction of taxonomic affiliations except in highly divergent phylogenetic clusters since mcrA has never been laterally transferred (as far as we know), there is a vast database of mcrA genes that have been taxonomically assigned, and methane-cycling Archaea are among the best physiologically and metagenomically characterized groups of anaerobic microorganisms. This sentence generally applies to many less-studied, less-conserved, and more frequently laterally transferred metabolic marker genes, but not necessarily to mcrA.

L. 91: again, unclear what is meant with "understudied". I suggest using a more objective and precise term, e.g. "physiologically unexplored" or simply "uncultivated".

Results:
L. 253-267: this literature review belongs into the Introduction, e.g a new paragraph inserted before the final paragraph of the Introduction.

Discussion:
This reads largely like a mixture of results and interpretation (discussion). I therefore suggest merging the Results and the Discussion into Results & Discussion, if this is allowed in PeerJ. Alternatively, all descriptive parts of the current Discussion can be separated from the more interpretive parts (e.g. concerning metabolic capabilities) and moved into the Results section.

Reviewer 2 ·

Basic reporting

The manuscript “Metabolic marker gene mining provides insight in global mcrA diversity and, coupled with targeted genome reconstruction, sheds light on metabolic versatility of the Methanomassiliicoccales” by Daan Speth and Victoria Orphan deals with the directed mining of public databases for mcrA genes in order to investigate the global importance of methanogenesis. Apart from some minor questions, I find the manuscript technically sound and the results justify the authors’ conclusions. The results are of substantial novelty, particularly, as the authors additionally recovered a previously unknown genome from the order of the Methanomassiliicocci and provide insight into its metabolism. Their findings highlight differences in previously described genomes and expand the known metabolic versatility of Methanomassiliicocci with the potential for aceticlastic methanogenesis. My major criticism is related to the citation of original literature in the introduction and the structuring of Results and Discussion section. I recommend a combined “Results and Discussion” section for this manuscript, if this is possible according to the journal style. Apart from these issues and the specific comments listed below I enjoyed reading this manuscript and congratulate the authors on this nice piece of research.

Experimental design

The experimental design is - apart from a few issues stated in the detailed comments below - state-of-the-art.

Validity of the findings

The data is robust. Particularly the fact, that the authors reconstructed full genomes to underpin their novel, gene-based findings is laudable.

Additional comments

Specific comments:
L 35: Please add a citation for genome-resolved metagenomics: Tyson et al., 2004 Nature.
L 38: This reference seems out of place; moreover, it is a review. Would be good to cite the original paper that performed metabolic predictions from reconstructed genomes from metagenomes (Tyson et al., 2004).
L 51: please specify if these genomes were redundant, i.e. from samples from the same environment. Would be good to give the number of different Methanomassiliicocci genomes.
L 59: Thomas et al. were not the first to describe binning of genomes. Other examples are Dick et al., 2009 in Genome Biology, in which scientists around Jill Banfield described k-mer based genome binning.
L 62: Delmont et al. were not the first to state that genome-resolved metagenomics necessitates manual curation. Please update citation.
L 58-66: The authors should also here (or elsewhere) discuss the disadvantage of their approach. For instance, the presented technique is limited to one specific gene and its diversity in public databases that provide the seed for the searches.
L 72-74: A citation for this statement is needed. Experience showed that genomes from metagenomes can be assembled and binned if they have at least an average coverage of ~7x, in some instance genomes with an average coverage of ~4x was successful. If a gene is recovered with a coverage <4x, how reliable is the reconstruction?
L 83: Scheller et al were probably not the first to describe that mcr gene products catalyze the last step in methanogenesis. Please provide a (better) reference here in addition to the citation for the initial step in ANMEs.
L 90: in L 86 the authors state that a substantial part of the diversity of mcr genes has been overlooked since multiple new organism groups have recently been discovered. So why do the authors restrict their analysis to the clade of Methanmassillicocci? That group is already known and has cultivated representatives. The point that I’m trying to make is, that the statement in L 86 might be a little bit too enthusiastic compared to the results that the authors got.
L 129: The authors state that it took 4 months to process the data. In the following sentence, they state how they sped up the process. How much time did it take after improving the scripts?
L 129: The authors only use the forward sequence for their analyses. However, they claim (L 72-74) that their process is able to detect functional genes of low abundance. This might actually not be possible by discarding 50% of the information available.
L 133: Why do the authors use DIAMOND? It is by far the fastest blast search tool when searching large datasets, but there are tools that use Hidden Markov Models to search read databases and they are tremendously fast. Example: “Short-Pair” in BMC Bioinformatics. 2017; 18(Suppl 12): 414. That process also uses paired-read information.
L 146: Why did the authors use IDBA_UD? Apart from the fact that it generally necessitates a tremendous amount of RAM, it is also very slow. The only advantage over very fast contigers like megahit is, that IDBA_UD is a scaffolder. But scaffolding is not useful for the approach the authors have chosen, since they specifically look for a fully assembled metabolic gene (mrcA). In contrast, the authors use Megahit for assembly of the draft genome, which could substantially benefit from scaffolding. It is recommended that the authors revisit the bioinformatics process as it is described in the paper.
L 147: Please indicate which mode you used for Prodigal (meta?).
L 150: Why were the mcrA genes of Syntrophoarchaeum not added in to the original database since it might have enhanced the detection of organisms related to those?
L 152: Please provide information on which model you have used for your ML tree. Was bootstrapping performed?
L 170: Please specify parameters for RAxML. Additionally, please justify the models that you used for the ML algorithm.
L 189-195: Please provide detailed information on the parameters used for each bioinformatics tool. Generally, the documentation of the different bioinformatics tools used in this manuscript is poorly described.
L 195: Please provide detailed information on how you curated the genome annotation.
L 196: please submit the genome to a public database. Your research benefits from public databases so it would be great to have your genome also in these databases for other researchers.
L 214: Here is the explanation for the question I asked for L 150. Maybe also explain in L 150?
L 202-219: This seems rather a discussion than a results section and it is also quite repetitive regarding the introduction.
L 265: How did you assess the environmental origin of your samples? Did you use a computational method?
L 283ff: The Discussion section rather seems to be a Results section since most of the text states new results on the MALP genome analysis.
L 348-349: please add citation for membrane complexes.
L 349-350: partial sentence, please rephrase.

Figure 1:
Please add commas to the value “12005048” to improve readability.

Figure 2:
Some sequences have zero sequences from PCR and zero sequences from metagenomes. How is this possible? Were they only represented by a genome sequence from an isolate? Please explain in the text.
C: Please specify abundance of sequences based on PCR. Is this from amplicon surveys? During Sanger times, people sometimes only submitted one representative sequence per cluster. Since I guess this has not been taken into account, the authors should spend a few more words on what this abundance actually means and what type of data is the basis for their analysis.

Reviewer 3 ·

Basic reporting

no comment

Experimental design

no comment.

Validity of the findings

no comment.

Additional comments

This work was to develop a gene-mining method to uncover the function-oriented unknown organisms from public metagenome database. The gene encoding the alpha subunit of methyl-coenzyme M reductase of methanogens was used as an example to illustrate how this new approach works. Over eighteen thousands of metagenomes available in public database were screened for the presence of mcrA and the obtained mcrA dataset was then used to assess the diversity and distribution of the Methanomassiliicoccales, the 7th order of methanogens which have remained largely unclear in terms of ecology and physiology. Furthermore, a draft genome related to the ‘Lake Pavin cluster’ within this group of methanogens was reconstructed. The data mining technique developed in this work appears very efficient. Firstly, mcrA sequences were obtained from the Pfam database and these were used to retrieve sequences available in NCBI-nr database. BLAST score method was used to eliminate the false positive hits. The mcrA dataset obtained had a much wider coverage to the known McrA sequences. Scripts were developed to perform DIAMOND search of mcrA sequences among over eighteen thousand metagenomes datasets. A stricter BSR method was used to screen the mcrA hits. Finally 1460 high quality near full length mcrA sequences were obtained. An in-depth research was conducted on the Methanomassiliicoccales-related mcrA sequences which showed the highest number of novel sequences retrieved from the multiple databases. A further effort was to reconstruct the genome of a Lake Pavin cluster Methanomassiliicoccales. The work was labor-demanding but well done and the technique developed shall be deemed to be very useful given the rocket growth of metagenome database. Following are a few minor comments.

Line 76, where not were.
Line 206, add ‘of’ between oxidation and methane.
Line 283 and below: it seem weird to separate Results and Discussion. Combination will be better.
Line 329, what do you mean ‘the second to last step of the methanogenic pathway’?
Lines 336-338, was methyltetrahydrofolate present in MALP?
Line 341, in line 336 it states that THMPT is likely absence. But here you thought THMPT can be a product of acetyl-CoA disproportionation?
Line 361, what other hydrogenases are predicted in MALP genome?
Line 408, delete ‘it’.

---

## Round 0.2 · accepted · Accept

Thank you for your careful and courteous responses to the reviewers.

#